# EXPLAINING BLACK BOX TEXT MODULES IN NATURAL LANGUAGE WITH LANGUAGE MODELS

## ABSTRACT

Large language models (LLMs) have demonstrated remarkable prediction performance for a growing array of tasks. However, their rapid proliferation and increasing opaqueness have created a growing need for interpretability. Here, we ask whether we can automatically obtain natural language explanations for black box text modules. A *text module* is any function that maps text to a scalar continuous value, such as a submodule within an LLM or a fitted model of a brain region. *Black box* indicates that we only have access to the module's inputs/outputs.

We introduce Summarize and Score (SASC), a method that takes in a text module and returns a natural language explanation of the module's selectivity along with a score for how reliable the explanation is. We study SASC in 3 contexts. First, we evaluate SASC on synthetic modules and find that it often recovers ground truth explanations. Second, we use SASC to explain modules found within a pre-trained BERT model, enabling inspection of the model's internals. Finally, we show that SASC can generate explanations for the response of individual fMRI voxels to language stimuli, with potential applications to fine-grained brain mapping. All code for using SASC and reproducing results is made available on Github.[1]

## 1 INTRODUCTION

Large language models (LLMs) have demonstrated remarkable predictive performance across a growing range of diverse tasks (Brown et al., 2020; Devlin et al., 2018). However, the inability to effectively interpret these models has led them to be characterized as black boxes. This opaqueness has debilitated their use in high-stakes applications such as medicine (Kornblith et al., 2022), and raised issues related to regulatory pressure (Goodman & Flaxman, 2016), safety (Amodei et al., 2016), and alignment (Gabriel, 2020). This lack of interpretability is particularly detrimental in scientific fields, such as neuroscience (Huth et al., 2016) or social science (Ziems et al., 2023), where trustworthy interpretation itself is the end goal.

To ameliorate these issues, we propose Summarize and Score (SASC). SASC produces *natural language explanations for text modules*. We define a *text module* $f$ as any function that maps text to a scalar continuous value, e.g. a neuron in a pre-trained LLM[2]. Given $f$, SASC returns a short natural language explanation describing what elicits the strongest response from $f$. SASC requires only black-box access to the module (it does not require access to the module internals) and no human intervention.

SASC uses two steps to ground explanations in the responses of $f$ (Fig. 1). In the first step, SASC derives explanation candidates by sorting $f$'s responses to ngrams and summarizing the top ngrams using a pre-trained LLM. In the second step, SASC evaluates each candidate explanation by generating synthetic text based on the explanation (again with a pre-trained LLM) and testing the response of $f$ to the text; these responses to synthetic text are used to assign an *explanation score* to each explanation, that rates the reliability of the explanation. Decomposing explanation into these separate steps helps mitigate issues with LLM hallucination when generating and evaluating explanations.

---

[1] Anonymized code available in supplementary zip file.

[2] Note that a neuron in an LLM typically returns a sequence-length vector rather than a scalar, so a transformation (e.g. averaging) is required before interpretation.

**(1) Summarize** ngrams into candidate explanations for module `f`

**(2) Score** candidate explanations

Figure 1: SASC pipeline for obtaining a natural language explanation given a module $f$. **(i)** SASC first generates candidate explanations (using a pre-trained LLM) based on the ngrams that elicit the most positive response from $f$. **(ii)** SASC then evaluates each candidate explanation by generating synthetic data based on the explanation and testing the response of $f$ to the data.

We evaluate SASC in two contexts. In our main evaluation, we evaluate SASC on synthetic modules and find that it can often recover ground truth explanations under different experimental conditions (Sec. 3). In our second evaluation, we use SASC to explain modules found within a pre-trained BERT model after applying dictionary learning (details in Sec. 4), and find that SASC explanations are often of comparable quality to human-given explanations (without the need for manual annotation). Furthermore, we verify that BERT modules which are useful for downstream text-classification tasks often yield explanations related to the task.

The recovered explanations yield interesting insights. Modules found within BERT respond to a variety of different phenomena, from individual words to broad, semantic concepts. Additionally, we apply SASC to modules that are trained to predict the response of individual brain regions to language stimuli, as measured by fMRI. We find that explanations for fMRI modules pertain more to social concepts (e.g. relationships and family) than BERT modules, suggesting possible different emphases between modules in BERT and in the brain. These explanations also provide fine-grained hypotheses about the selectivity of different brain regions to semantic concepts.

## 2 METHOD

SASC aims to interpret a text module $f$, which maps text to a scalar continuous value. For example $f$ could be the output probability for a single token in an LLM, or the output of a single neuron extracted from a vector of LLM activations. SASC returns a short explanation describing what elicits the strongest response from $f$, along with an *explanation score*, which rates how reliable the explanation is. In the process of explanation, SASC uses a pre-trained *helper LLM* to perform summarization and to generate synthetic text. To mitigate potential hallucination introduced by the helper LLM, SASC decomposes the explanation process into 2 steps (Fig. 1) that greatly simplify the task performed by the helper LLM:

**Step 1: Summarization** The first step generates candidate explanations by summarizing ngrams. All unique ngrams are extracted from a pre-specified corpus of text and fed through the module $f$. The ngrams that elicit the largest positive response from $f$ are then fed through the helper LLM for summarization. To avoid over-reliance on the very top ngrams, we select a random subset of the top ngrams in the summarization step. This step is similar to prior works which summarize ngrams using manual inspection/parse trees (Kádár et al., 2017; Na et al., 2019), but the use of the helper LLM enables flexible, automated summarization.

The computational bottleneck of SASC is computing $f$'s response to the corpus ngrams. This computation requires two choices: the corpus underlying the extracted ngrams, and the length of ngrams to extract. Using a larger corpus/higher order ngrams can make SASC more accurate, but the computational cost grows linearly with the unique number of ngrams in the corpus. The corpus should

be large enough to include relevant ngrams, as the corpus limits what generated explanations are possible (e.g. it is difficult to recover mathematical explanations from a corpus that contains no math). To speed up computation, ngrams can be subsampled from the corpus.

**Step 2: Synthetic scoring**  The second step aims to evaluate each candidate explanation and select the most reliable one. SASC generates synthetic data based on each candidate explanation, again using the helper LLM. Intuitively, if the explanation accurately describes $f$, then $f$ should output large values for text related to the explanation (*Text*$^+$) compared to unrelated synthetic text (*Text*$^-$).[3] We then compute the explanation score as follows:

$$\text{Explanation score} = \mathbb{E}[f(\textit{Text}^+) - f(\textit{Text}^-)] \text{ with units } \sigma_f, \tag{1}$$

where a larger score corresponds to a more reliable explanation. We report the score in units of $\sigma_f$, the standard deviation of $f$'s response to the corpus. An explanation score of $1\sigma_f$ means that synthetic text related to the explanation increased the mean module response by one standard deviation compared to unrelated text. SASC returns the candidate explanation that maximizes this difference, along with the synthetic data score. The selection of the highest-scoring explanation is similar to the reranking step used in some prompting methods, e.g. (Shin et al., 2020), but differs in that it maximizes $f$'s response to synthetic data rather than optimizing the likelihood of a pre-specified dataset.

**Limitations and hyperparameter settings**  While effective, the explanation pipeline described here has some clear limitations. First and foremost, SASC assumes that $f$ can be concisely described in a natural language string. This excludes complex functions or modules that respond to a non-coherent set of inputs. Second, SASC only describes the inputs that elicit the largest responses from $f$, rather than its full behavior. Finally, SASC requires that the pre-trained LLM can faithfully perform its required tasks (summarization and generation). If an LLM is unable to perform these tasks sufficiently well, users may treat the output of SASC as candidate explanations to be vetted by a human, rather than final explanations to be used.

We use GPT-3 (`text-davinci-003`, Feb. 2023) (Brown et al., 2020) as the helper LLM (see LLM prompts in Appendix A.2). In the summarization step, we use word-level trigrams, choose 30 random ngrams from the top 50 and generate 5 candidate explanations. In the synthetic scoring step, we generate 20 synthetic strings (each is a sentence) for each candidate explanation, half of which are related to the explanation.

## 3  RECOVERING GROUND TRUTH EXPLANATIONS FOR SYNTHETIC MODULES

This section describes our main evaluation of SASC: its ability to recover explanations for synthetic modules with a known ground truth explanation.

**Experimental setup for synthetic modules**  We construct 54 synthetic modules based on the pre-trained Instructor embedding model (Su et al., 2022) (`hkunlp/instructor-xl`). Each module is based on a dataset from a recent diverse collection (Zhong et al., 2021; 2022) that admits a simple, verifiable keyphrase description describing each underlying dataset, e.g. *related to math* (full details in Table A2). Each module is constructed to return high values for text related to the module's groundtruth keyphrase and low values otherwise. Specifically, the module computes the Instructor embedding for an input text and for the groundtruth keyphrase; it then returns the negative Euclidean distance between the embeddings. We find that the synthetic modules reliably produce large values for text related to the desired keyphrase (Fig. A3).

We test SASC's ability to recover accurate explanations for each of our 54 modules in 3 settings: (1) The *Default* setting extracts ngrams for summarization from the dataset corresponding to each module, which contains relevant ngrams for the ground truth explanation. (2) The *Restricted corpus* setting checks the impact of the underlying corpus on the performance of SASC. To do so, we restrict

---

[3]The unrelated synthetic text should be neutral text that omits the relevant explanation, but may introduce bias into the scoring if the helper LLM improperly generates negative synthetic texts. Instead of synthetic texts, a large set of neutral texts may be used for *Text*$^-$, e.g. samples from a generic corpus.

Table 1: Explanation recovery performance. For both metrics, higher is better. Each value is averaged over 54 modules and 3 random seeds; errors show standard error of the mean.

| | **SASC** | | Baseline (ngram summarization) | |
|---|---|---|---|---|
| | Accuracy | BERT Score | Accuracy | BERT Score |
| Default | 0.883 ±0.03 | 0.712 ±0.02 | 0.753 ±0.02 | 0.622 ±0.05 |
| Restricted corpus | 0.667 ±0.04 | 0.639 ±0.02 | 0.540 ±0.02 | 0.554 ±0.05 |
| Noisy module | 0.679 ±0.04 | 0.669 ±0.02 | 0.456 ±0.02 | 0.565 ±0.06 |
| Average | **0.743** | **0.673** | 0.582 | 0.580 |

Table 2: Explanation recovery accuracy when varying hyperparameters for the *Default* setting; averaged over 54 modules and 3 random seeds.

| | SASC (Original) | SASC (Bigrams) | SASC (4-grams) | SASC (LLaMA-2 summarizer) | SASC (LLaMA-2 generator) | Baseline (Gradient based) | Baseline (Topic modeling) |
|---|---|---|---|---|---|---|---|
| Acc. | 0.883±0.03 | 0.815±0.04 | 0.889±0.03 | 0.870±0.03 | 0.852±0.04 | 0.093±0.01 | 0.111±0.01 |
| BERT Score | 0.712±0.02 | 0.690±0.03 | 0.714±0.02 | 0.705±0.02 | 0.701±0.02 | 0.351±0.01 | 0.388±0.01 |

Table 3: Examples of recovered explanations for different modules in the *Default* setting.

| | Groundtruth Explanation | SASC Explanation |
|---|---|---|
| Correct | atheistic | atheism and related topics, such as theism, religious beliefs, and atheists |
| | environmentalism | environmentalism and climate action |
| | crime | crime and criminal activity |
| | sports | sports |
| | definition | defining or explaining something |
| | facts | information or knowledge |
| Incorrect | derogatory | negative language and criticism |
| | ungrammatical | language |
| | subjective | art and expression |

the ngrams we use for generating explanation candidates to a corpus from a random dataset among the 54, potentially containing less relevant ngrams. (3) The *Noisy module* setting adds Gaussian noise with standard deviation $3\sigma_f$ to all module responses in the summarization step.

**Baselines and evaluation metrics** We compare SASC to three baselines: (1) ngram-summarization, which summarizes top ngrams with an LLM, but does not use explanation scores to select among candidate explanations (essentially SASC without the scoring step); (2) gradient-based explanations (Poerner et al., 2018), which use the gradients of $f$ with respect to the input to generate maximally activating inputs; (3) topic modeling (Blei et al., 2003), which learns a 100-component dictionary over ngrams using latent dirichlet allocation.

We evaluate similarity of the recovered explanation and the groundtruth explanation in two ways: (1) Accuracy: verifying whether the ground truth is essentially equivalent to the recovered explanation via manual inspection and (2) BERT-score (Zhang et al., 2019)[4]. We find that these two metrics, when averaged over the datasets studied here, have a perfect rank correlation, i.e. every increase in average accuracy corresponds to an increase in average BERT score. For topic modeling, accuracy is evaluated by taking the top-30 scoring ngrams for the module (as is done with SASC), finding the 5 topics with the highest scores for these ngrams, and manually checking whether there is a match between the groundtruth and any of the top-5 words in any of these topics.

**SASC can recover ground truth descriptions** Table 1 shows the performance of SASC at recovering ground truth explanations. In the *Default* setting, SASC successfully identifies 88% of the ground truth explanations. In the two noisy settings, SASC still manages to recover explanations

---

[4]BERT-score is calculated with the base model `microsoft/deberta-xlarge-mnli` (He et al., 2021).

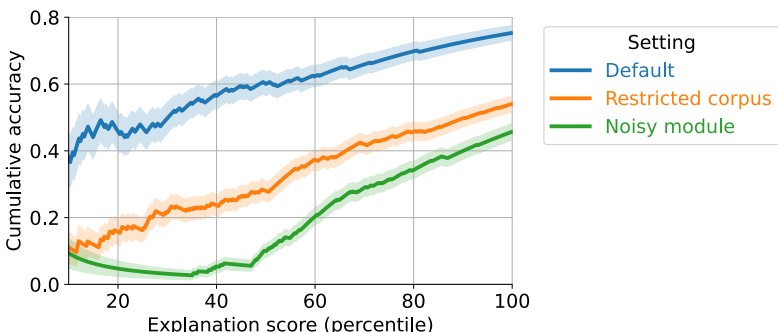

Figure 2: Cumulative accuracy at recovering the ground truth explanation increases as a function of explanation score. Error bars show standard error of the mean.

67% and 68% of the time for the *Restricted ngrams* and *Noisy module* settings, respectively. In all cases, SASC outperforms the ngram-summarization baseline.

Table 2 shows the results for the *Default* setting when varying different modeling choices. Performance is similar across various choices, such as using bigrams or 4-grams rather than trigrams in the summarization step, or when using the LLaMA-2 13-billion parameter model (Touvron et al., 2023b) as the helper LLM rather than GPT-3. Additionally, we find that explanation performance increases with the capabilities of the helper LLM used for summarization/generation (Fig. A1). Table 2 also shows that the gradient-based baseline fails to accurately identify the underlying groundtruth text, consistent with previous work in prompting (Singh et al., 2022b; Shin et al., 2020) and that topic modeling performs poorly, largely because the topic model fails to construct topics relevant to each specific module, as the same input ngrams are shared across all modules.

Table 3 shows examples of correct and incorrect recovered explanations along with the ground truth explanation. For some modules, SASC finds perfect keyword matches, e.g. *sports*, or slight paraphrases, e.g. *definition → defining or explaining something*. For the incorrect examples, the generated explanation is often similar to the ground truth explanation, e.g. *derogatory → negative language and criticism*, but occasionally, SASC fails to correctly identify the underlying pattern, e.g. *ungrammatical → language*. Some failures may be due to the inability of ngrams to capture the underlying explanation, whereas others may be due to the constructed module imperfectly representing the ground truth explanation.

Fig. 2 shows the cumulative accuracy at recovering the ground truth explanation as a function of the explanation score. Across all settings, accuracy increases as a function of explanation score, suggesting that higher explanation scores indicate more reliable explanations. This also helps validate that the helper LLM is able to sucessfully generate useful synthetic texts for evaluation.

## 4    GENERATING EXPLANATIONS FOR BERT TRANSFORMER FACTORS

Next, we evaluate SASC using explanations for modules within BERT (Devlin et al., 2018) (bert-base-uncased). In the absence of ground truth explanations, we evaluated the explanations by (i) comparing them to human-given explanations and (ii) checking their relevance to downstream tasks.

**BERT transformer factor modules**    One can interpret any module within BERT, e.g. a single neuron or an expert in an MOE (Fedus et al., 2022); here, we choose to interpret *transformer factors*, following a previous study that suggests that they are amenable to interpretation (Yun et al., 2021). Transformer factors learn a transformation of activations across layers via dictionary learning (details in Appendix A.3; corpus used is the WikiText dataset (Merity et al., 2016)). Each transformer factor is a module that takes as input a text sequence and yields a scalar dictionary coefficient, after averaging over the input's sequence length. There are 1,500 factors, and their coefficients vary for each of BERT's 13 encoding layers.

**Comparison to human-given explanations**    Table 4 compares SASC explanations to those given by humans in prior work (31 unique explanations from Table 1, Table 3 and Appendix in (Yun et al., 2021)). They are sometimes similar with different phrasings, e.g. *leaving or being left* versus

Table 4: Comparing sample SASC to human-labeled explanations for BERT transformer factors. Win percentage shows how often the SASC explanation yields a higher explanation score than the human explanation. See all explanations and scores in Table A4.

| SASC Explanation | Human Explanation |
|---|---|
| names of parks | Word "park". Noun. a common first and last name. |
| leaving or being left | Word "left". Verb. leaving, exiting |
| specific dates or months | Consecutive years, used in football season naming. |
| idea of wrongdoing or illegal activity | something unfortunate happened. |
| introduction of something new | Doing something again, or making something new again. |
| versions or translations | repetitive structure detector. |
| publishing, media, or awards | Institution with abbreviation. |
| names of places, people, or things | Unit exchange with parentheses |
| SASC win percentage: **61%** | Human explanation win percentage: 39% |
| SASC mean explanation score: **1.6$\sigma_f$** | Human explanation mean explanation score: 1.0$\sigma_f$ |

*Word "left"*, and sometimes quite different, e.g. *publishing, media, or awards* versus *Institution with abbreviation*. For each transformer factor, we compare the explanation scores for SASC and the human-given explanations. The SASC explanation score is higher 61% of the time and SASC's mean explanation score is $1.6\sigma_f$ compared to $1.0\sigma_f$ for the human explanation. This evaluation suggests that the SASC explanations can be of similar quality to the human explanations, despite requiring no manual effort.

**Mapping explained modules to text-classification tasks**   We now investigate whether the learned SASC explanations are useful for informing which downstream tasks a module is useful for. Given a classification dataset where the input $X$ is a list of $n$ strings and the output $y$ is a list of $n$ class labels, we first convert $X$ to a matrix of transformer factor coefficients $X_{TF} \in \mathbb{R}^{n \times 19,500}$, where each row contains the concatenated factor coefficients across layers. We then fit a sparse logistic regression model to $(X_{TF}, y)$, and analyze the explanations for the factors with the 25 largest coefficients across all classes. Ideally, these explanations would be relevant to the text-classification task; we evaluate what fraction of the 25 explanations are relevant for each task via manual inspection.

We study 3 widely used text-classification datasets: *emotion* (Saravia et al., 2018) (classifying tweet emotion as sadness, joy, love, anger, fear or surprise), *ag-news* (Zhang et al., 2015) (classifying news headlines as world, sports, business, or sci/tech), and *SST2* (Socher et al., 2013) (classifying movie review sentiment as positive or negative). Table 5 shows results evaluating the BERT transformer factor modules selected by a sparse linear model fit to these datasets. A large fraction of the explanations for selected modules are, in fact, relevant to their usage in downstream tasks, ranging from 0.35 for *Emotion* to 0.96 for *AG News*. The *AG News* task has a particularly large fraction of relevant explanations, with many explanations corresponding very directly to class labels, e.g. *professional sports teams → sports* or *financial investments → business*. See the full set of generated explanations in Appendix A.3.

**Patterns in SASC explanations**   SASC provides 1,500 explanations for transformer factors in 13 layers of BERT. Fig. 3 shows that the explanation score decreases with increasing layer depth, suggesting that SASC better explains factors at lower layers. The mean explanation score across all layers is $1.77\sigma_f$.

To understand the breakdown of topics present in the explanations, we fit a topic model (with Latent Dirichlet Allocation (Blei et al., 2003)) to the remaining explanations. The topic model has 10 topics and preprocesses each explanation by converting it to a vector of word counts. We exclude all factors that do not attain an explanation score of at least $1\sigma_f$ from the topic model, as they are less likely to be correct. Fig. 4 shows each topic along with the proportion of modules whose largest topic coefficient is for that topic. Topics span a wide range of categories, from syntactic concepts (e.g. *word, end, ..., noun*) to more semantic concepts (e.g. *sports, physical, activity, ...*).

Table 5: BERT modules selected by a sparse linear model fit to text-classification tasks. First row shows the fraction of explanations for the selected modules which are relevant to the downstream task. Second row shows test accuracy for the fitted linear models. Bottom section shows sample explanations for modules selected by the linear model which are relevant to the downstream task. Values are averaged over 3 random linear model fits (error bars show the standard error of the mean).

|  | Emotion | AG News | SST2 |
|---|---|---|---|
| Fraction relevant | $0.35\pm_{0.082}$ | $0.96\pm_{0.033}$ | $0.44\pm_{0.086}$ |
| Test accuracy | $0.75\pm_{0.001}$ | $0.81\pm_{0.001}$ | $0.84\pm_{0.001}$ |
| *Sample relevant explanations* | negative emotions such as hatred, disgust, disdain, rage, and horror | people, places, or things related to japan | a negative statement, usually in the form of not or nor |
|  | injury or impairment | professional sports teams | hatred and violence |
|  | humor | geography | harm, injury, or damage |
|  | romance | financial investments | something being incorrect or wrong |

## 5 GENERATING EXPLANATIONS FOR FMRI-VOXEL MODULES

**fMRI voxel modules** A central challenge in neuroscience is understanding how and where semantic concepts are represented in the brain. To meet this challenge, one line of study predicts the response of different brain voxels (i.e. small regions in the brain) to natural language stimuli (Huth et al., 2016; Jain & Huth, 2018). We analyze data from (LeBel et al., 2022) and (Tang et al., 2023), which consists of fMRI responses for 3 human subjects as they listen to 20+ hours of narrative stories from podcasts. We fit modules to predict the fMRI response in each voxel from the text that the subject was hearing by extracting text embeddings with a pre-trained LLaMA model (`decapoda-research/llama-30b-hf`) (Touvron et al., 2023a). After fitting the modules on the training split and evaluating them on the test split using bootstrapped ridge regression, we generate SASC explanations for 1,500 well-predicted voxel modules, distributed evenly among the three human subjects and diverse cortical areas (see details on the fMRI experimental setup in Appendix A.4.1).

**Voxel explanations** Table 6 shows examples of explanations for individual voxels, along with three top ngrams used to derive the explanation. Each explanation unifies fairly different ngrams under a common theme, e.g. *sliced cucumber, cut the apples, sauteed shiitake...* → *food preparation*. In some cases, the explanations recover language concepts similar to known selectivity in sensory modalities, e.g. face selectivity in IFSFP (Tsao et al., 2008) and selectivity for non-speech sounds such as laughter in primary auditory cortex (Hamilton et al., 2021). The ngrams also provide more fine-grained hypotheses for selectivity (e.g. *physical injury or pain*) compared to the coarse semantic categories proposed in earlier language studies (e.g. *emotion* (Huth et al., 2016; Binder et al., 2009; Mitchell et al., 2008)).

Fig. 4 shows the topics that fMRI explanations best fit into compared with BERT transformer factors. The proportions for many topics are similar, but the fMRI explanations yield a much greater proportion for the topic consisting of social words (e.g. *relationships*, *communication*, *family*) and perceptual words (e.g. *action*, *movement*, *physical*). This is consistent with prior knowledge, as the largest axis of variation for fMRI voxels is known to separate social concepts from physical concepts (Huth et al., 2016).

The selected 1,500 voxels often achieve explanation scores considerably greater than zero for their explanations (mean explanation score $1.27\sigma_f \pm 0.029$). Fig. 3 (bottom) shows the mean explanation score for the six most common fMRI regions of interest (ROIs) among the voxels we study here. Across regions, the fMRI voxel modules generally attain explanation scores that are slightly lower than BERT modules in early layers and slightly higher than BERT modules in the final layers.

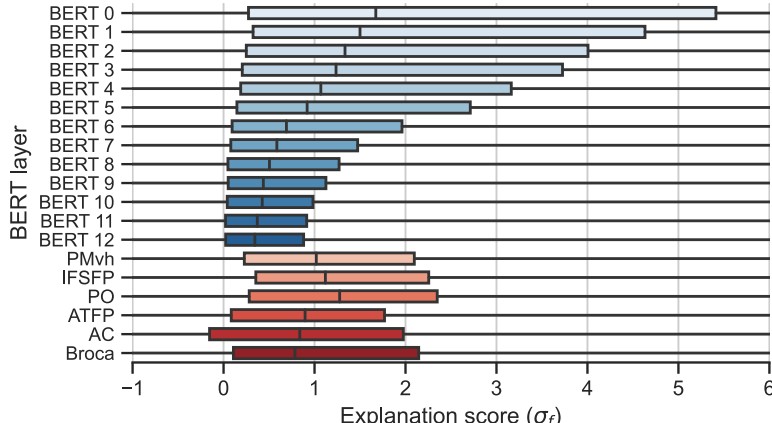

Figure 3: Explanation score for BERT (blue) and fMRI (orange) modules. As the BERT layer increases, the explanation score tends to decrease, implying modules are harder to explain with SASC. Across regions, explanation scores for fMRI voxel modules are generally lower than scores for BERT modules in early layers and comparable to scores for the final layers. Boxes show the median and interquartile range. ROI abbreviations: premotor ventral hand area (PMvh), anterior temporal face patch (ATFP), auditory cortex (AC), parietal operculum (PO), inferior frontal sulcus face patch (IFSFP), Broca's area (Broca).

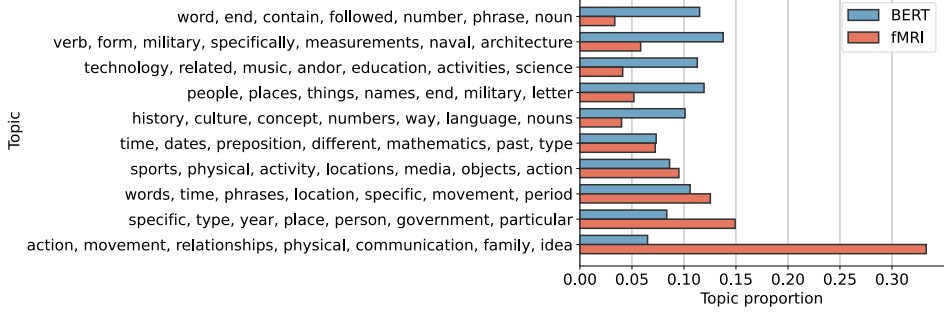

Figure 4: Topics found by LDA for explanations of BERT factors and fMRI voxels. Topic proportion is calculated by assigning each explanation to the topic with the largest coefficient. Topic proportions for BERT/fMRI explanations largely overlap, although the bottom topic consisting of physical/social words is much more prevalent in fMRI explanations.

We also find some evidence that the generated fMRI voxel explanations can explain not just the fitted module, but also brain responses to unseen data (see Appendix A.4.2). This suggests that the voxel explanations here can serve as hypotheses for followup experiments to affirm the fine-grained selectivity of specific brain voxels.

## 6 RELATED WORK

Table 6: Examples of recovered explanations for individual fMRI voxel modules. All achieve an fMRI predicted correlation greater than 0.3 and an explanation score of at least $1\sigma$. The third column shows 3 of the ngrams used to derive the explanation in the SASC summarization step.

| Explanation | ROI | Example top ngrams |
|---|---|---|
| looking or staring in some way | IFSFP | eyed her suspiciously, wink at, locks eyes with |
| relationships and loss | ATFP | girlfriend now ex, lost my husband, was a miscarriage |
| physical injury or pain | Broca | infections and gangrene, pulled a muscle, burned the skin |
| counting or measuring time | PMvh | count down and, weeks became months, three more seconds |
| food preparation | ATFP | sliced cucumber, cut the apples, sauteed shiitake |
| laughter or amusement | ATFP, AC | started to laugh, funny guy, chuckled and |

**Explaining modules in natural language** A few related works study generating natural language explanations. MILAN (Hernandez et al., 2022) uses patch-level information of visual features to generate descriptions of neuron behavior in vision models. iPrompt (Singh et al., 2022b) uses automated prompt engineering and D5 (Zhong et al., 2023; 2022)/GSClip (Zhu et al., 2022) use LLMs to describe patterns in a dataset (as opposed to describing a module, as we study here). In concurrent work, (Bills et al., 2023) propose an algorithm similar to SASC that explains individual neurons in an LLM by predicting token-level neuron activations.

Two very related works use top-activating ngrams/sentences to construct explanations: (1) (Kádár et al., 2017) builds an explanation by *manually* inspecting the top ngrams eliciting the largest module responses from a corpus using an omission-based approach. (2) (Na et al., 2019) similarly extracts the top sentences from a corpus, but summarizes them using a parse tree. Alternatively, (Poerner et al., 2018) use a gradient-based method to generate maximally activating text inputs.

**Explaining neural-network predictions** Most prior works have focused on the problem of explaining a *single prediction* with natural language, rather than an entire module, e.g. for text classification (Camburu et al., 2018; Rajani et al., 2019; Narang et al., 2020), or computer vision (Hendricks et al., 2016; Zellers et al., 2019). Besides natural language explanations, some works explain individual prediction via feature importances (e.g. LIME (Ribeiro et al., 2016)/SHAP (Lundberg et al., 2019)), feature-interaction importances (Morris et al., 2023; Singh et al., 2019; Tsang et al., 2017), or extractive rationales (Zaidan & Eisner, 2008; Sha et al., 2021). They are not directly comparable to SASC, as they work at the prediction-level and do not produce a natural-language explanation.

**Explaining neural-network representations** We build on a long line of recent work that explains neural-network *representations*, e.g. via probing (Conneau et al., 2018; Liu & Avci, 2019), via visualization (Zeiler & Fergus, 2014; Karpathy et al., 2015), by categorizing neurons into categories (Bau et al., 2017; 2018; 2020; Dalvi et al., 2019; Gurnee et al., 2023), localizing knowledge in an LLM (Meng et al., 2022; Dai et al., 2021), or distilling information into a transparent model (Tan et al., 2018; Ha et al., 2021; Singh et al., 2022a).

**Natural language representations in fMRI** Using the representations from LLMs to help predict brain responses to natural language has become common among neuroscientists studying language processing in recent years (Jain & Huth, 2018; Wehbe et al., 2014; Schrimpf et al., 2021; Toneva & Wehbe, 2019; Antonello et al., 2021; Goldstein et al., 2022). This paradigm of using "encoding models" (Wu et al., 2006) to better understand how the brain processes language has been applied to help understand the cortical organization of language timescales (Jain et al., 2020; Chen et al., 2023), examine the relationship between visual and semantic information in the brain (Popham et al., 2021), and explore to what extent syntax, semantics or discourse drives brain activity (Caucheteux et al., 2021; Kauf et al., 2023; Reddy & Wehbe, 2020; Pasquiou et al., 2023; Aw & Toneva, 2022; Kumar et al., 2022; Oota et al., 2022; Tuckute et al., 2023).

## 7 DISCUSSION

SASC could potentially enable much better mechanistic interpretability for LLMs, allowing for automated analysis of submodules present in LLMs (e.g. attention heads, transformer factors, or experts in an MOE), along with an explanation score that helps inform when an explanation is reliable. Trustworthy explanations could help audit increasingly powerful LLMs for undesired behavior or improve the distillation of smaller task-specific modules. SASC also could also be a useful tool in many scientific pipelines. The fMRI analysis performed here generates many explanations which can be directly tested via followup fMRI experiments to understand the fine-grained selectivity of brain regions. SASC could also be used to generate explanations in a variety of domains, such as analysis of text models in computational social science or in medicine.

While effective, SASC has many limitations. SASC only explains a module's top responses, but it could be extended to explain the entirety of the module's responses (e.g. by selecting top ngrams differently). Additionally, due to its reliance on ngrams, SASC fails to capture low-level text patterns or patterns requiring long context, e.g. patterns based on position in a sequence. Future explanations could consider adding information beyond ngrams, and also probe the relationships between different modules to explain circuits of modules rather than modules in isolation.

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

## A  APPENDIX

### A.1  METHODOLOGY DETAILS EXTENDED

Table A1: Statistics on corpuses used for explanation. Wikitext is used for BERT explanation and Moth stories are used for fMRI voxel explanation.

|  | Unique unigrams | Unique bigrams | Unique trigrams |
|---|---|---|---|
| Wikitext (Merity et al., 2016) | 157k | 3,719k | 9,228k |
| Moth stories (LeBel et al., 2022) | 117k | 79k | 140k |
| Combined | 158k | 3,750k | 9,334k |

**Prompts used in SASC**   The summarization step summarizes 30 randomly chosen ngrams from the top 50 and generates 5 candidate explanations using the prompt *Here is a list of phrases:\n{phrases}\nWhat is a common theme among these phrases?\nThe common theme among these phrases is ___.*

In the synthetic scoring step, we generate similar synthetic strings with the prompt *Generate 10 phrases that are similar to the concept of {explanation}:.* For dissimilar synthetic strings we use the prompt *Generate 10 phrases that are not similar to the concept of {explanation}:.* Minor automatic processing is applied to LLM outputs, e.g. parsing a bulleted list, converting to lowercase, and removing extra whitespaces.

