# OpenReview forum: "Explaining black box text modules in natural language with language models"
_ICLR.cc/2024/Conference — Submitted to ICLR 2024_

### Official Review · Reviewer_CSdo · 2023-10-30

**Soundness:** 2 fair
**Presentation:** 2 fair
**Contribution:** 3 good
**Rating:** 6
**Confidence:** 3

**Summary:**

This paper introduces Summarize and Score (SASC), a method which generates natural language descriptions of text models, where a text model is any function that takes in text and outputs a scalar. This method does not require the model to be differentiable, and can operate by accessing paired input/outputs of the model.

I've read both the main paper and the supplemental. Their proposed model consists of three steps:
1. The probe step. Where they score natural language phrases taken from dataset.
2. The summary step. They take a random subset of the top-scoring phrases, and provide them to a language model. The language model is prompted to provide multiple explanation of what activates a language model.
2. The scoring step. When they prompt a language model to generate phrases related and unrelated to the explanation. An average of the difference is computed after proving the generate phrases to the original probed model, which is used to rank the proposed explainations.

They evaluate their proposed method in a couple of ways:
1. Using the INSTRUCTOR-XL text embedding models, with a prompt for the model to cluster. Followed by negative euclidean distance.
2. Using BERT combined with non-negative factorized dictionary coefficients.
3. For fMRI voxels on a passive listening task, on "The Moth" among other works.

**Strengths:**

I think on balance the authors did a good exploration of their method, and the method is generally well described.

1. By evaluating on embedding models, you have a ground truth target to evaluate against. Assuming the embedding model is accurate, which is not an unreasonable assumption.
2. By interpreting BERT transformer factors against previous human evaluations -- this is a direct comparison against a human baseline.
3. By looking at fMRI data, this evaluates SASC on a noisy scenario where the various brain regions are can only be observed under a noisy setting.

The assumptions used in the paper are weak and reasonable, and do not require differentiability assumptions, a residual/skip connection assumption (for example in the logit lens paper). Compared to "Language models can explain neurons in language models" paper from OpenAI, their approach of using a corpus rather than per-token scoring is more sound, and it allows for more context dependent explanations.

Given that interpretability is a significant issue in the use of large language models, this paper addresses a timely and important problem.

**Weaknesses:**

I have a couple of concerns on the evaluation done in this paper.

1. I think their corpus + score approach is more sound than the OpenAI's approach of token-wise scoring. But there is one problem I would like to see resolved:

   a. There is no ablation study on the corpus size and the effect on explanation scores. Ideally the authors could explore the outcome using for example 10k, 20k, 40k, 80k, .... all n-grams.

2. The scoring step in my view is suspect, and introduces an unnecessary confound in terms of their procedures. It seems very wrong to evaluate $E(f(\text{Text}^{+})-f(\text{Text}^{-}))$. In this case, you are asking the language model to generate the phrases unrelated to an explanation, so the final score is not just related to how good the positives are, but also how "good" the negatives are. Perhaps a more correct step should be $E(f(\text{Text}^{+})-f(\text{Text}^{\text{all}}))$. This is less problematic if this was just used as an internal component, but I think it is problematic when the authors use it as a point of comparison in Table 4, Figure 3, Table 6, fMRI experiments, Table A4, and Table A8.

3. The "Default" setting in "synthetic modules" experiment is not very sound in my view. Ideally the exploration of the 54 modules should not be relying on a corpus which is known to contain the relevant ngrams, but the exploration of the modules should use the same (random) selection of ngrams.

4. It is a bit questionable that the authors use manual inspection to evaluate their method. Scanning both the main text and the supplemental, the authors do not use Amazon Mechanical Turk or Prolific, and do not include any details on this human study. I can only assume that this experiment was done by the authors themselves, which is prone to bias. A more accurate approach would be to take a different text embedding model (for example bge-large, other models are fine too), compute the n-way cosine similarities among the 54 models to the 54 explainations, and compute the argmax agreement (is the n-th explanation generated by the SASC cosine similarity maximized by the n-th model), or even the cosine distance when n-way classification is not appropriate.

5. The clarity in some specific methods are poor. There is no details in either the main text or the supplemental. Concretely, I was unable to find details for the following:

    a. ngram summarization baseline used in Table 1. I looked at the references provided by the author (Kadar et al. and Na et al.), neither are actually performing ngram summarization. The first citation uses a omission based approach, while the second uses a parsing + text replication approach.

   b . Can the authors describe how they get the spearman rank correlation sem? I would not otherwise ask if the effect is strong, but the correlation is very low. Do the authors find the sem via a bootstrap? Could the authors report a t-statistic instead?

6. It seems like this proposed method would only work on semantic selectivity, and it is unclear if this method can work on low-level text patterns. This is okay, but the authors should more clearly discuss this limitation in their paper.

Overall I like the paper. And the proposed method is well described, but the author's evaluation broadly does not strike me as sound. I would happily re-evaluate if the authors can answer my questions.

**Questions:**

Please see above.

---

> ### Author Response · Authors · 2023-11-16
> **Thank you for your comments**
>
> | 1. Corpus-size ablation
>
> Thanks for this suggestion, we have added a new Fig A2 which uses the WikiText corpus (which contains 9,822k unique trigrams) and randomly subsamples its ngrams. Performance improves sharply from 10k to 50k and plateaus around 100k.
>
> | 2. Scoring step, E[f(Text+)-f(Text-)] vs E[f(Text+)-f(Text^all).
>
> Thanks, we agree using Text- can add more noise to the procedure and have added this note in the methods:
>
> “The unrelated synthetic text should be neutral text that omits the relevant explanation, but may introduce bias into the scoring if the helper LLM improperly generates negative synthetic texts. Instead of synthetic texts, a large set of neutral texts may be used for Text−, e.g. samples from a generic corpus.”
>
> Additionally we find that the empirical difference between using Text- and Text^All is minimal. Specifically, we add Fig A5, which shows results for different modules when evaluated on E[f(Text+)], E[f(Text+)-f(Text-)]. It also shows E[f(Text^All)] (the off-diagonal elements of Fig A5 left subpanel). We find that in this setting, there is essentially no difference between E[f(Text^All)] and E[f(Text-)] (the difference between the two is 0.02$\sigma_f \pm 0.1\sigma_f$).
>
> As LLMs improve, we do believe E[f(Text-)] will be a better choice than E[f(Text^All)], as Text- can correctly omit samples that should be in Text+, but otherwise stay neutral.
>
> | 3. Modules should use the same ngrams.
>
> See response to (1), where we include this as a new experiment (Fig A2), using the WikiText corpus as the shared corpus across modules. We find that the performance using WikiText falls slightly below the corpus-specific “Default” setting; the biggest errors are on dataset 15 and 16, which are both specific to “Hillary Clinton” – this is likely because the subsampled ngrams from WikiText are missing ngrams related to these tasks.
>
>  We do still include the synthetic modules “Default” setting, as in many settings (e.g. the fMRI setting), practitioners can use their domain knowledge to select a corpus that is relevant to the modules/explanations at hand.
>
> | 4. Replacing manual inspection with text-embedding model
>
> We have taken additional steps to mitigate the potential bias of our manual inspection:
>
> First and foremost, we have added BERT score, a more objective similarity measure (similar to the cosine distance you mention), along with manual-inspection accuracy in Table 1 and Table 2. We find that the two metrics, when averaged over the 54 explanations, have a perfect rank correlation, i.e. every increase in average accuracy corresponds to an increase in average BERT score. Note that we compute BERT scores using the fairly capable deberta model recommended by the BERT-score repo (microsoft/deberta-xlarge-mnli).
>
> We also add Fig A4 showing the similarities computed in the Default setting using the bge-large model as you suggest. A clear diagonal pattern is visible, as one would expect based on the other evaluations. Taking the 54-class classification accuracy using the argmax on these similarities results in an accuracy of 81.5%, lower than the manual inspection accuracy of 88.3%. This is not too surprising as many datasets have very similar groundtruth explanations, e.g. math/statistics, and a human may mark both as accurate. When we compute the top-2 accuracy of the bge-large similarities, the accuracy rises to 88.9%, now slightly higher than the manual inspection accuracy.
>
> We also add more information on the manual inspection, showing manual classifications in Table A6-A8 and showing regexes that guide the manual inspection in Table A3.
>
> | 5. Method clarity
>
> Apologies for the confusion. We have cleaned up some of the writing; in particular, we have added a “Baselines and evaluation metrics” header in Section 3. The first paragraph of this section describes the baseline.
>
> Indeed, neither Kadar et al. nor Na et al. perform ngram summarization, but these references are the closest we can find to giving natural-language explanations of black-box text modules by summarizing ngrams; we updated their description in the related work:
>
> | 5b . Spearman rank sem
>
> The sem (Sec A.4.2) is computed across the rank correlation for the 1,500 voxels, so it is simply the standard deviation of the 1500 correlations (one for each voxel) divided by $\sqrt{1500}$. This can be translated into a t-statistic by simply dividing the mean by this sem, yielding $t=0.033/0.002=16.5$.
>
> Nevertheless, we agree the correlation is quite low when making predictions as described in A.4.2; in a followup study we are seeing much stronger results when testing these explanations on unseen data in a more natural way (which requires followup fMRI experiments).
>
> | 6. Limiting to semantic selectivity
>
> We agree and have added this sentence to the Discussion: “Additionally, due to its reliance on ngrams, SASC fails to capture low-level text patterns or patterns requiring long context, e.g. patterns based on position in a sequence”

---

> > ### Comment · Reviewer_CSdo · 2023-11-17
> > **Quick request**
> >
> > Thanks for the response.
> >
> > Could you change the text color of your revisions so it is more clear what you've changed?

---

> > > ### Author Response · Authors · 2023-11-18
> > >
> > > We have added a revisions.pdf file in the Supplement with the highlighted changes (note that the biggest changes are in the appendix).

---

> > > > ### Comment · Reviewer_CSdo · 2023-11-18
> > > > **Score raised**
> > > >
> > > > Thank you for the additional experiments.
> > > >
> > > > I've raised my score to a 6, I think on balance the paper is interesting and would be a contribution to ICLR.
> > > >
> > > > I still believe that the original paper should have been revised with the more sound experimental setup, rather than those revisions being in the supplemental. But given the time constraints it is understandable.

---

### Official Review · Reviewer_Tacm · 2023-10-30

**Soundness:** 3 good
**Presentation:** 3 good
**Contribution:** 3 good
**Rating:** 8
**Confidence:** 4

**Summary:**

The paper introduces Summarize and Score (SASC), a method to automatically generate natural language explanations for black box text processing modules, such as those in large language models. SASC was evaluated on synthetic modules, BERT modules, and fMRI data mapping text stimuli to voxel responses. The method often recovered ground truth explanations on synthetic data. When applied to BERT, SASC enabled inspection of the model's internals, while application to fMRI data allowed fine-grained mapping of language selectivity in the brain. Limitations include only explaining top module responses in isolation rather than relationships between modules. Overall, SASC shows promise for interpretability of LLMs and could be a useful scientific tool for domains analyzing text models like social science, medicine, and neuroscience.

**Strengths:**

1. This paper tackles the timely and important topic of interpreting black box natural language processing models like large language models. The proposed SASC framework enables inspecting model internals and explaining model predictions in an interpretable way.
2. The SASC framework is novel and theoretically grounded. The approach of using a separate helper model to generate explanations is creative.
3. The experiments reveal fascinating patterns in how different models generate explanations. The results indicate SASC has potential for analyzing text models in fields like social science, medicine, and neuroscience.

**Weaknesses:**

Please refer to the questions section

**Questions:**

1. How do the authors decide whether an explanation is correct or not? For example, for Table 3, why ‘facts’ is considered to be equal for ‘information or knowledge’, while ‘language’ is considered to be unequal to ‘ungrammatical’? What is the criteria to decide if an explanation is correct or not?
2. It would be great if the authors could give some hints in scenarios where the proposed SASC algorithm does not have satisfactory results. Whether the reason is due to the limitation of the ngrams or the limitations of the helper LLMs.
3. It would be great if the authors could clarify in more detail what does ‘transformer factors’ mean for BERT in Section 4. Additionally, what are the ground truth for these transformer factors? How is the SASC win percentage: 61% calculated?
4. What GPT-3 is used as the helper LLM, rather than a more advanced model, such as GPT-3.5?
5. Another interesting finding in Figure 3 is that, as the BERT layer increases, the explanation score tends to decrease. Could the authors give some hints on why this happens?

---

> ### Author Response · Authors · 2023-11-16
> **Thank you for your comments**
>
> Thank you for your thoughtful comments.
>
> | 1. How do the authors decide whether an explanation is correct or not? For example, for Table 3, why ‘facts’ is considered to be equal for ‘information or knowledge’, while ‘language’ is considered to be unequal to ‘ungrammatical’? What is the criteria to decide if an explanation is correct or not?
>
> The criteria is judgement by manual inspection. As this may be subject to bias, we have added BERT scores to Table 2 in addition to Table 1 and explicitly show the judgements made (e.g. Table A6, A7, A8). We find that the average BERT scores have a perfect rank correlation with average accuracy, i.e. every increase in average accuracy also yields an increase in average BERT score. We have added this quote to the methods section to clarify this:
>
> “We evaluate similarity of the recovered explanation and the groundtruth explanation in two ways: (1) Accuracy: verifying whether the ground truth is essentially equivalent to the recovered explanation via manual inspection and (2) BERT-score (Zhang et al., 2019)4. We find that these two metrics, when averaged over the datasets studied here, have a perfect rank correlation, i.e. every increase in average accuracy corresponds to an increase in average BERT score.”
>
> | 2. It would be great if the authors could give some hints in scenarios where the proposed SASC algorithm does not have satisfactory results. Whether the reason is due to the limitation of the ngrams or the limitations of the helper LLMs.
>
> Empirically, we find that when using GPT-3, the helper LLM is rarely the source of explanation errors (smaller LLMs do introduce errors, see Fig A1). Rather, the source of error is usually SASC’s reliance on ngrams. This is the case for explanations that may require long context (e.g. a module that responds to subjectivity), or that depend on positional information. We have added this comment to the Discussion to clarify this: “Additionally, due to its reliance on ngrams, SASC fails to capture low-level text patterns or patterns requiring long context, e.g. patterns based on position in a sequence.”
>
>
> | 3. It would be great if the authors could clarify in more detail what does ‘transformer factors’ mean for BERT in Section 4. Additionally, what are the ground truth for these transformer factors? How is the SASC win percentage: 61% calculated?
>
> We have added some more details on transformer factors to the main text and section A.3. Here are the relevant quotes from the main text:
>
> Transformer factors: “we choose to interpret transformer factors, following a previous study that suggests that they are amenable to interpretation (Yun et al., 2021). Transformer factors learn a transformation of activations across layers via dictionary learning (details in Appendix A.3; corpus used is the WikiText dataset (Merity et al., 2016)). Each transformer factor is a module that takes as input a text sequence and yields a scalar dictionary coefficient, after averaging over the input’s sequence length. There are 1,500 factors, and their coefficients vary for each of BERT’s 13 encoding layers.”
> Groundtruth: “In the absence of ground truth explanations, we evaluated the explanations by (i) comparing them to human-given explanations and (ii) checking their relevance to downstream tasks.”
> Win percentage: “Win percentage shows how often the SASC explanation yields a higher explanation score than the human explanation.”
>
> | 4. What GPT-3 is used as the helper LLM, rather than a more advanced model, such as GPT-3.5?
>
> Thanks for this suggestion. We have a limited budget and GPT-3 was cheap and sufficiently effective for this task. GPT-3 was also easier to compare to smaller models, such as text-babbage-001 & text-curie-001 (Fig A1). Before camera ready, we will re-run the main experiments with the best available closed-source model (GPT-4 or better).
>
> | 5. Another interesting finding in Figure 3 is that, as the BERT layer increases, the explanation score tends to decrease. Could the authors give some hints on why this happens?
>
> Indeed this is interesting. We find that this is a consistent trend across interpretability works where higher layers tend to represent more abstract concepts which are harder to succinctly explain.

---

### Official Review · Reviewer_beYn · 2023-11-01

**Soundness:** 3 good
**Presentation:** 2 fair
**Contribution:** 2 fair
**Rating:** 5
**Confidence:** 4

**Summary:**

This paper proposes a method called Summarize and Score (SASC) that generates natural language explanations for black-box text modules. The authors evaluate SASC on synthetic modules, explaining modules found within a pre-trained BERT model, and generating explanations for the response of individual fMRI voxels to language stimuli. The results show that SASC can generate high-quality explanations that are both accurate and human-readable. The authors also provide insights into the inner workings of the pre-trained BERT model and demonstrate the potential of SASC for fine-grained brain mapping. Overall, the paper presents a promising approach for improving the interpretability of black box text modules.

**Strengths:**

1.	The paper proposes a new method called Summarize and Score (SASC) that generates natural language explanations for black-box text modules.
2.	The authors evaluate SASC on synthetic modules, explaining modules found within a pre-trained BERT model and generating explanations for the response of individual fMRI voxels to language stimuli.
3.	The results show that SASC can generate high-quality explanations that are both accurate and human-readable.
4.	The proposed method has significant implications for improving the interpretability of machine learning models.

**Weaknesses:**

1.	The paper does not compare SASC to other state-of-the-art methods for generating natural language explanations for black box models. It would be useful to compare SASC to other methods, such as LIME and SHAP, to determine how it performs in comparison.
2.	In SASC design, the first step is to input the n-grams from the reference corpus into the text module, which could contain many instances. How to ensure the efficiency of the proposed method remains undiscussed. It would be better for authors to provide a detailed analysis of the computational complexity of SASC. It would be useful to provide information on the computational requirements of the proposed method and potential ways to optimize it.
3.	The design of synthetic scoring is heuristic and without any theoretical analysis to support this design. And most importantly, there is no ablation study to verify the effectiveness of this model design.

**Questions:**

1.	Can you provide a comparison of SASC to other state-of-the-art methods for generating natural language explanations for black box models? This would help readers understand how SASC performs in comparison to other methods.
2.	Can you provide more information on the computational complexity of SASC? Specifically, how long does it take to generate explanations for a given module and what are the computational requirements of the proposed method? This would help readers understand the practical limitations of SASC and potential ways to optimize it.

---

> ### Author Response · Authors · 2023-11-16
> **Thank you for your comments**
>
> Thank you for your thoughtful comments.
>
> | 1. The paper does not compare SASC to other state-of-the-art methods for generating natural language explanations for black box models. It would be useful to compare SASC to other methods, such as LIME and SHAP, to determine how it performs in comparison.
>
> Apologies for any confusion caused by the writing. SASC and post-hoc interpretation methods such as LIME/SHAP have different goals:
> |             | SASC                                                 | LIME/SHAP                                                  |
> | ----------- | ---------------------------------------------------- | ---------------------------------------------------------- |
> | Output      | Natural-language explanation                         | Feature importance scores                                  |
> | Granularity | Module                                               | Individual prediction                                      |
> | Goal        | Describe what elicits strongest response from module | Describe contributions of different inputs to a prediction |
>
> They are useful in different scenarios. If one wants to compare them directly, we can take the highest-scoring inputs for each prediction and then try to summarize them into natural language. This is what our ngram summarization baseline in Table 2 aims to do, and we find that it performs decently, but not as well as SASC.
>
> We have edited the writing to clarify this point in the first paragraph of section 2 and added this note in the related work section:
>
>
> “Besides natural language explanations, some works explain individual prediction via feature importances (e.g. LIME (Ribeiro et al., 2016)/SHAP (Lundberg et al., 2019)), feature-interaction importances (Morris et al., 2023; Singh et al., 2019; Tsang et al., 2017), or extractive rationales (Zaidan & Eisner, 2008; Sha et al., 2021). These works are not directly comparable to SASC, as they work at the prediction-level and do not produce a natural-language explanation.”
>
>
> | 2. In SASC design, the first step is to input the n-grams from the reference corpus into the text module, which could contain many instances. How to ensure the efficiency of the proposed method remains undiscussed. It would be better for authors to provide a detailed analysis of the computational complexity of SASC. It would be useful to provide information on the computational requirements of the proposed method and potential ways to optimize it.
>
> Thanks for this comment. We have added this paragraph on computational complexity as the third paragraph in the methods section:
>
> “The computational bottleneck of SASC is computing f ’s response to the corpus ngrams. This computation requires two choices: the corpus underlying the extracted ngrams, and the length of ngrams to extract. Using a larger corpus/higher order ngrams can make SASC more accurate, but the computational cost grows linearly with the unique number of ngrams in the corpus. The corpus should be large enough to include relevant ngrams, as the corpus limits what generated explanations are possible (e.g. it is difficult to recover mathematical explanations from a corpus that contains no math). To speed up computation, ngrams can be subsampled from the corpus.”
>
> Additionally, we have added new experiments in Fig A2 showing computational tradeoffs between the number of ngrams and the performance; we find that increasing the number of ngrams beyond 100k no longer improves performance.
>
> | 3. The design of synthetic scoring is heuristic and without any theoretical analysis to support this design. And most importantly, there is no ablation study to verify the effectiveness of this model design.
>
> Indeed it is difficult to develop rigorous theory for an LLM-based method like SASC. As a result, we do perform ablations in Table 1 and Fig 2 showing the importance of the synthetic scoring: Table 1 compares SASC to ngram summarization without synthetic scoring, finding that the synthetic scoring yields an improvement of >16% accuracy on average. Moreover, Fig 2 shows that accuracy at recovering the ground truth explanation increases as a function of the synthetic score. Thus, selecting the best explanation based on synthetic score improves performance.
>
> We have made these ablations clearer in the updated text (see the new “Baselines and evaluation metrics paragraph” in Section 3).

---

### Meta-Review · Area_Chair_XAwg · 2023-12-05

**Metareview:**

The authors propose SASC, a method that can summarize a module (either some part of a LM or some model of a brain region) with a natural language description of the selectivity and reliability. Evaluations show that this approach can summarize various modules - including synthetic settings with known ground truth, as well as more real world settings for BERT and fMRI.

Strengths: reasonable experiment design, better overall setup than OpenAI's describing LMs with language work.

Weaknesses: the original human eval design was pretty fundamentally flawed (see comments by CSdo), and there's broader questions of justifying this type of NL interpretability by reviewer beYn

**Justification For Why Not Higher Score:**

The paper is very much borderline, but reviewer CSdo points out some very good points, and some of the experiments (like the human evals) probably should be redone - this would best be done via a resubmission to another conference.

**Justification For Why Not Lower Score:**

N/A

---

### Decision · Program_Chairs · 2024-01-16

Reject